# Combination of ultra-rapid DNA purification (PURE) and loop-mediated isothermal amplification (LAMP) for rapid detection of *Trypanosoma cruzi* DNA in dried blood spots

**Silvia A. Longhi**[1]\*, **Lady J. García Casares**[1], **Arturo A. Muñoz-Calderón**[1], **Julio Alonso-Padilla**[2,3], **Alejandro G. Schijman**[1]\*

**1** Laboratorio de Biología Molecular de la Enfermedad de Chagas, Instituto de Investigaciones en Ingeniería Genética y Biología Molecular "Dr Héctor Torres", INGEBI-CONICET, Buenos Aires, Argentina, **2** Barcelona Institute for Global Health (ISGlobal), Hospital Clinic—University of Barcelona, Barcelona, Spain, **3** CIBER de Enfermedades Infecciosas, Instituto de Salud Carlos III (CIBERINFEC, ISCIII), Madrid, Spain

\* slonghi@ingebi-conicet.gov.ar (SAL); aschijman@ingebi-conicet.gov.ar (AGS)

## Abstract

### Background

Chagas disease or American trypanosomiasis, a neglected tropical disease, is a persistent Public Health problem in Latin America and other, non-endemic, countries. Point-of-care (POC) sensitive methods are still needed to improve and extend early diagnosis in acute infections such as congenital Chagas disease. The objective of this study was to analytically evaluate in the lab the performance of a qualitative POC molecular test (Loop-mediated isothermal amplification (LAMP), Eiken, Japan) for rapid diagnosis of congenital Chagas disease employing FTA cards or Whatman 903 filter paper as solid supports for small-scale volumes of human blood.

### Methodology/principal findings

We used human blood samples artificially infected with cultured *T. cruzi* strains to assess the analytical performance of the test in comparison with liquid blood anticoagulated with heparin. The DNA extraction process was evaluated using the ultrarapid purification system PURE manufactured by Eiken Chemical Company (Tokio, Japan) over artificially infected liquid blood or different amounts of dried blood spot (DBS) 3- and 6-mm pieces of FTA and Whatman 903 paper. LAMP was performed on a AccuBlock (LabNet, USA) heater or in the Loopamp LF-160 incubator (Eiken, Japan), and visualization of results was either done at naked eye, using the LF-160 device or P51 Molecular Fluorescence Viewer (minipcr bio, USA). Best conditions tested showed a limit of detection (LoD) with 95% accuracy (19/20 replicates) of 5 and 20 parasites/mL, respectively for heparinized fluid blood or DBS samples. FTA cards showed better specificity than Whatman 903 filter paper.

**Data Availability Statement:** All relevant data are within the manuscript and its Supporting Information files.

**Funding:** This study was funded by the Global Health Innovative Technology Fund (GHIT Fund, G2020-203; Tokyo, Japan) to JAP and AGS. We would also like to acknowledge the co-funding by Fundación Mundo Sano, and the support from PAHO through the Small Grant Programme (TDR Chagas Project LEG ID 39002) to AGS, ERANet LAC Joint Call 2016-2017 Cochaco to AGS, and PICT 2020-0862 from the Ministry of Science and Technology, Argentina to AGS and SAL. The funders had no role in study design, data collection and analysis, decision to publish, or preparation of the manuscript.

**Competing interests:** The authors have declared that no competing interests exist.

## Conclusions/significance

Procedures to operate LAMP reactions from small volumes of fluid blood or DBS in FTA were standardized for LAMP detection of *T. cruzi* DNA. Our results encourage prospective studies in neonates born to seropositive women or oral Chagas disease outbreaks to operationally evaluate the method in the field.

### Author summary

Chagas disease is a global health problem in endemic and non-endemic regions due to migration. Early diagnosis of newborns to seropositive mothers is of utmost importance to Public Health, enabling prompt access to highly effective anti-parasitic treatment. For this, it is essential to develop rapid and sensitive methods at point-of-care (POC). The aim of this work was to estimate the analytical performance of the molecular amplification technique named Loop-mediated isothermal amplification (LAMP) as POC strategy to detect *Trypanosoma cruzi* in small volumes of blood collected in dried blood spots. The procedure was carried out in human blood samples spiked with *T. cruzi* strains on Whatman 903 or FTA commercial filter papers in comparison with fluid blood in heparin as anticoagulant. We used an ultra-rapid DNA extraction method, tested the LAMP reaction in two different heaters and visualized the results either by the naked eye or using fluorescence viewers. FTA cards showed excellent sensitivity and specificity; LAMP could be performed in a simple heater and results easily visualized with fluorescence viewers. The entire process entails handling basic laboratory devices that will enable POC diagnosis of congenital infection and other acute cases of Chagas disease such as those derived from oral outbreaks.

## Introduction

Chagas disease, caused by the parasite *Trypanosoma cruzi*, is a neglected infectious disease that affects between six and eight million people in the Americas. Current estimates indicate that there are roughly 28,000 new acute cases each year, and nearly 65 million people live at risk of contracting the disease by vector-borne transmission, blood or congenital transmission, or food-borne transmission. It has been recognized by the Pan-American Health Organization (PAHO) that there are substantial needs in terms of increasing access, coverage, and quality of care within national health care systems, mainly in primary care networks [1].

Considering the very high sensitivity of molecular methods for the detection of acute *T. cruzi* infections, their availability and adoption would be extremely useful for diagnosing cases emerging in oral outbreaks, detecting infection relapse in immunocompromised patients, or for the early detection of congenitally transmitted Chagas disease from *T. cruzi* seropositive mothers to their newborns [1,2].

Indeed, Loop mediated isothermal amplification (LAMP), is a molecular approach suitable to resource-limited laboratories, because the strand-displacement-*Bst* DNA polymerase works at 60–65°C and does not require the use of a thermocycler, but only a thermal block or dry heater [3,4]. In addition, the procedure is much simpler to run than currently available polymerase chain reaction (PCR) methods, and the visualisation of the amplification product can

be done by the naked eye, using blue LED lightning or followed in real time by turbidity or fluorescence.

Several LAMP procedures have already been proposed for the detection of *T. cruzi* infections [5–10]. The most studied of all has been the prototype kit developed by Eiken Chemical Company (Tokyo, Japan) based on *T. cruzi* nuclear satellite DNA (satDNA) sequences and containing dried reagents on the inside of the microtube caps, which has proven as sensitive as real-time quantitative PCR (qPCR) in blood samples collected from patients with acute, congenital and Chagas disease reactivations, as well as in patients under treatment monitoring [5–8].

An ultra-rapid DNA extraction method named PURE (Eiken), designed for the performance of point-of-care (POC) molecular methods could represent an adequate procedure for POC diagnosis of congenital infection with *T. cruzi*, starting from very low volumes of blood or from dried blood spots (DBS) in filter paper, such as Whatman FTA cards or 903 protein saver card (GE Healthcare, UK) [11]. This combination of DBS, PURE, and LAMP has already been proposed for the detection of malaria [11]. Similarly, such an approach could be of extreme relevance to guarantee coverage for the early diagnosis of congenital Chagas disease in vast rural areas of highly endemic regions where many deliveries occur in primary health centers that do not count with equipped laboratories or even at the mothers´ domiciles. Upon collection, DBS can be stored and transported at room temperature, a major advantage to take them to the corresponding regional laboratory where LAMP could be performed.

This work aimed to standardize and assess the analytical performance of the combination of PURE-DNA extraction followed by LAMP to detect *T. cruzi* in FTA cards and Whatman 903 filter paper in comparison to fluid blood.

## Methods

### Ethics statement

Written informed consent was obtained from the healthy donors before blood collection, in accordance with current Argentine legislation (Blood Donation Law No. 22990, Res. No. 1409/15).

### Study design

In order to establish the protocol that provides better analytical sensitivity, different conditions of sampling, incubation and visualization were tested (Fig 1). An ultra-rapid DNA extraction method was chosen to analyze 30 to 50 µL of heparinized blood, one to 12 pieces of 3-mm or one or two 6-mm dried blood in FTA cards, and one or two DBS in Whatman 903 filter paper. In addition to improve sensitivity, two incubation times for the LAMP reaction were tested. Also, two brands of incubators and three visualization methods were examined to have more equipment options according to the facilities of the health centres. After analysing all conditions, a step-by-step protocol was set-up (S1 Table).

### Parasite strains and preparation of spiked blood samples

We used stocks from Sylvio X-10 (*T. cruzi* genotype I, TcI or Discrete Typing Unit I (DTU-I)), Y (TcII or DTU-II) and CL Brener (TcVI or DTU-VI) that are representative strains of different number of copies of satDNA [12]. Heparin-treated seronegative human blood samples were spiked with axenic cultures of epimastigote forms of Sylvio X-10, Y or CL-Brener stocks. Briefly, from an axenic culture of epimastigote forms of known concentration, serial dilutions were made to obtain between 1–10 parasites in 0.5 µL. These dilutions were counted in Neubauer chamber under a 40x inverted microscopy objective and transferred to tubes containing

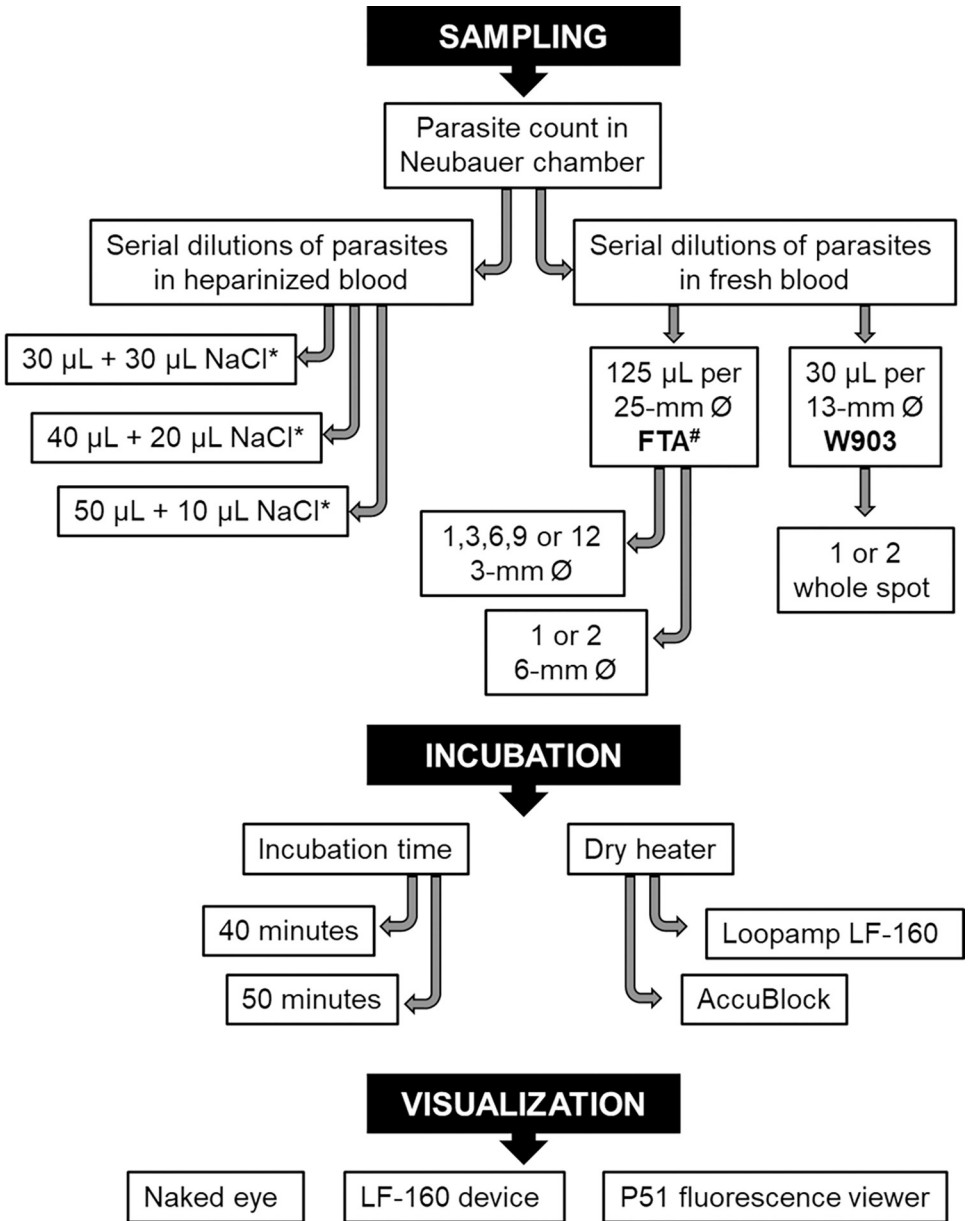

**Fig 1. Study design.** Flow chart showing the different conditions tested for the PURE-LAMP procedure. *NaCl final concentration in the heating tube was 10 mM. #DBSs in FTA cards were tested with or without Whatman FTA Purification Reagent washes.

1 mL of heparin-treated blood in order to exactly obtain a concentration of 1000 parasites/mL (H1000). Then, dilutions of 50, 20, 15, 10 and 5 parasites/mL in heparin-treated blood or in non-coagulated fresh blood were prepared from that original H1000 sample. In order to avoid blood clotting during the preparation process of the suspensions and seeding in fresh blood, each blood suspension was kept in a dry bath at 37°C. Then, 30 μL or 125 μL of each suspension were immediately seeded in Whatman 903 (GE Healthcare, Cardiff, UK) or QIAcard Flinders Technology Associates (FTA) Classic cards (Qiagen, UK), respectively. Blood spots were left at room temperature at least 4 hours to completely dry (and thus become DBS) before

storage in individual zip lock bags. These were kept at room temperature and fluid blood samples were kept frozen at -20°C until use.

A suspension of *Leishmania mexicana* in heparinized blood was also prepared as described above to obtain a final concentration of $1x10^5$ parasites/mL (H100,000). Then, dilutions of 5000 parasites/mL in heparin-treated blood or in non-coagulated fresh blood to seed FTA cards were prepared from that original H100,000 stock.

### DNA extraction using PURE system

Regarding fluid blood samples anticoagulated with heparin, we tested the DNA purification of 30, 40 and 50 μL of blood.

The PURE procedure was done following the manufacturer´s instructions, with some modifications, as follows: (i)30 μL of 334 mM NaCl plus 30 μL of blood were added into the heating tube containing extraction buffer, provided by the kit, mixed by inverting three to five times, and placed in a heating block at 75°C for 5 min; (ii) the heating tube was attached to the adsorbent tube, also provided, and the powder contained into the adsorbent tube was completely mixed with the solution; and (iii) the DNA was directly eluted into the LAMP micro-tube using the injection cap provided by the PURE kit.

When higher volumes of fluid blood were tested (40 and 50 μL), 20 and 10 μL of NaCl were added respectively, at the appropriate concentrations to reach the same 10 mM NaCl final concentration in the heating tube.

PURE DNA extraction of DBS in Whatman 903 involved testing one or two whole spots, cut down to small pieces with sterile scissors.

For DBS in FTA, comparison of DNA extractions were carried out in whole circles with or without a first washing step using an FTA specific solution (Whatman FTA Purification Reagent, GE Healthcare, UK) prior to PURE extraction. In addition, different numbers of 3-mm and 6-mm spots were punched out using a sterilized metal puncher (Uni-Core, GE Healthcare, UK).

For all DBS conditions, 30 μL of a 334 mM NaCl solution were added into the heating tube and we followed the steps for DNA extraction described for fluid blood.

### LAMP amplification

LAMP was carried out following the recommendations of the manufacturer. Briefly, approximately 30 μL from the PURE purified DNA eluate were loaded on each LAMP micro-tube so that the space delimited by the two marks of the LAMP micro-tubes was covered. Then, sample-loaded LAMP micro-tubes were incubated 40 or 50 min at 65°C for amplification reaction followed by 5 min at 80°C for enzyme inactivation. Each run included negative and positive controls supplied in the kit.

For reading out the results, presence (positive) or absence (negative) of fluorescent light was addressed using the LF-160 viewer (Eiken, Japan), the P51 viewer (minipcr bio, Cambridge, MA, USA) or simply by naked eye (Fig 2).

### Inclusivity and exclusivity

To evaluate inclusivity of LAMP starting from spiked fluid and DBS samples processed by PURE, samples containing 20 and 50 parasite equivalents/mL of Sylvio X-10 (Tc I), Y (Tc II) and CL-Brener (Tc VI) stocks were tested in duplicate.

To evaluate exclusivity of LAMP, 5000 parasite equivalents per mL (par.Eq./mL) of *Leishmania. mexicana* spiked in heparinised blood and in DBS samples were tested in duplicate.

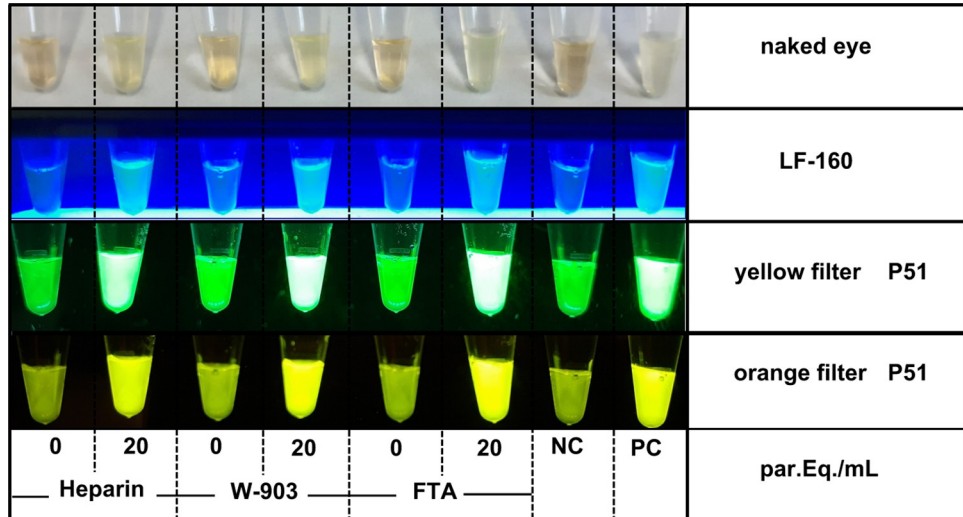

**Fig 2. Visualization of LAMP results.** From top to bottom: visualization by naked eye; by using the LF-160 device; and the P51 fluorescence viewer with a yellow acrylic sheet (above) and an orange acrylic sheet (below). Positive blood samples contain 20 par. Eq./ mL of Sylvio X-10 stock on the indicated support. NC and PC are negative and positive controls, respectively, provided in the kit.

## Analytical sensitivity and specificity

In order to explore the parasite concentration around the limit of detection at 95% accuracy (LoD95), serial dilutions of fluid blood and DBS spiked with *T. cruzi* cells were tested in triplicate. As Y and CL-Brener strains have more similar satDNA dosage between them than with respect to Sylvio X-10, experiments were done using CL-Brener and Sylvio X-10 spiked samples [12]. Next, for the lowest concentration rendering positive LAMP findings on triplicates, at least 20 further replicates were tested, following the guidelines for analytical validation of molecular methods published on 16th March 2020 during the COVID-19 pandemics [13]. Moreover, blood samples without parasites were included for each of the tested supports, as specificity controls.

## Reproducibility

Parasite concentrations close to the LoD95 were evaluated in FTA cards and spiked heparin blood using two different heaters: Loopamp LF-160 and AccuBlock. Blood in Whatman 903 was not tested, because this type of sampling format showed specificity issues.

## Stability of DBS in FTA cards

FTA DBS specimens spiked with Sylvio X-10 epimastigote forms were stored at room temperature in an individual zip-sealed plastic bag and protected from light, for over one, two, six, 12 and 18 months after preparation until PURE DNA extraction and LAMP evaluation.

## Results

### Selection of best conditions of PURE-LAMP testing for each type of sample

Artificial samples containing serial dilutions of cultured parasites were used to test different experimental conditions, aiming to select the better combinations of the PURE DNA

**Table 1. Limit of detection of LAMP in fluid blood samples or DBS containing blood spiked with parasite stocks from two different DTUs.**

| Type of sample | Sampling conditions | *T. cruzi* stock | *T. cruzi* concentration | 40 min LAMP | | 50 min LAMP | |
|---|---|---|---|---|---|---|---|
| | | | | Pos/Tested | Positivity | Pos/Tested | Positivity |
| HEPARIN BLOOD | 30 µL | | 0 par.Eq./mL | 0/20 | 0% | 0/20 | 0% |
| | | Sylvio X-10 | 10 par.Eq./mL | 25/26 | 96% | 20/20 | 100% |
| | | CL-Brener | 5 par.Eq./mL | 15/20 | 75% | 19/20 | 95% |
| WHATMAN 903 | 2 DBS | | 0 par.Eq./mL | 5/20 | 25% | 8/20 | 40% |
| | | Sylvio X-10 | 15 par.Eq./mL | 17/20 | 85% | 19/20 | 95% |
| | | CL-Brener | 10 par.Eq./mL | 17/20 | 85% | 18/20 | 90% |
| FTA | 6-mm disc | | 0 par.Eq./mL | 0/20 | 0% | 0/20 | 0% |
| | | Sylvio X-10 | 20 par.Eq./mL | 26/28 | 93% | 20/20 | 100% |
| | | CL-Brener | 15 par.Eq./mL | 19/21 | 91% | 20/21 | 95% |

**Whatman 903**: each DBS contains 30 µL of dried blood in Whatman 903 filter paper; **FTA**: each DBS was blotted with 125 µL blood in FTA cards and a single 6 mm punched disc was used per test; **par. Eq./mL**: parasite equivalent per millilitre; **min**: minutes; **Pos/Tested**: number of positive results out of tested samples.

extraction followed by LAMP testing in heparinized blood and DBS (either on Whatman 903 or FTA) samples:

Heparinized blood: different starting volumes of fluid blood (30 to 50 µL) were tested. Overall, 30 µL of blood mixed with the same volume of a 334 mM NaCl solution provided in the kit meant the starting sample volume recommended by the manufacturer. In an attempt to increase the sensitivity, 40 and 50 µL of blood plus the appropriate volume and concentration of NaCl recommended by the manufacturer were tested. Nevertheless, the increase in sample volume did not result in a better sensitivity. Contrary to what was expected, an absence of amplification was obtained even at high concentrations of parasites per mL, indicating that the capacity limit of this system to remove contaminants and inhibitors from the blood is up to 30 µL of sample. Thus, the best results were obtained using 30 µL of blood plus 30 µL of 334 mM NaCl, so this starting volume was chosen for the following experiments.

Whatman 903 spots: each DBS contained 30 µL of dried blood. One or two DBS samples were submitted to PURE preparation and LAMP assay. Two DBS samples improved positivity, so this condition was employed to determine analytical sensitivity (Table 1).

FTA cards: one hundred and twenty-five µL of blood were added per circle. No differences were obtained in DBS with or without prior washing. Moreover, one, three, six, nine and 12 punches of 3 mm or one and two 6 mm punches were tested. Better sensitivity was obtained using one or two 6-mm punches than using higher numbers of 3-mm punches. A single 6-mm punch was then chosen for subsequent experiments.

## Inclusivity and exclusivity

The PURE-LAMP assay was tested using DNA extracted by PURE from blood spiked with three different *T. cruzi* strains, namely Silvio X-10 (Tc DTU I), Y strain (Tc DTU II) and CL-Brener (Tc DTU VI).

Fig 3 shows the results obtained using DNA from 6-mm punches of spiked blood loaded in FTA cards and incubated in Loopamp LF-160 or AccuBlock heaters and visualized by the naked eye or using the blue LED light of the LF160.

No differences were obtained irrespectively of using as incubator for the LAMP reaction the AccuBlock dry heater or the LF-160 device. However, once photographed, the results were clearer visualized using LED light to illuminate the LAMP micro-tubes contains than when naked eye visualization was used (Fig 3B).

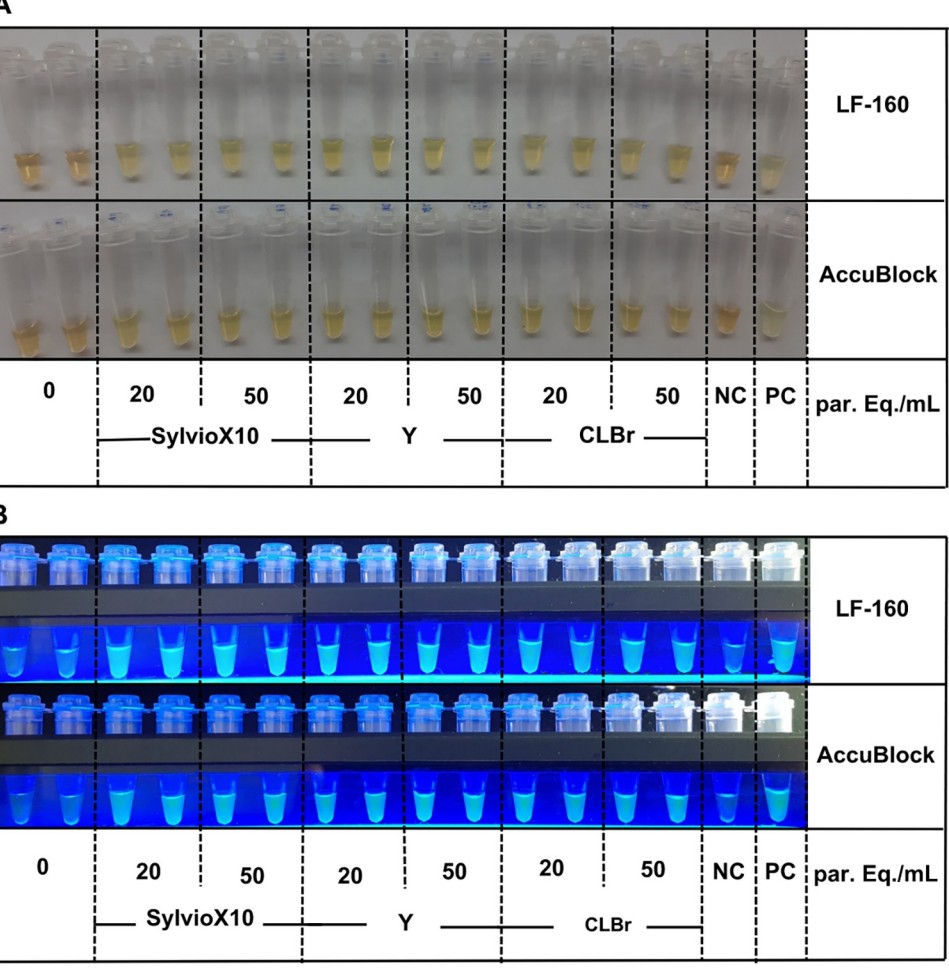

**Fig 3. Inclusivity and reproducibility of LAMP using FTA cards seeded with three different *T. cruzi* strains in two different incubators and different ways of visualization.** (A) LAMP results obtained after incubation in LF-160 or AccuBlock and naked eye visualization. (B) LAMP results obtained after incubation in LF-160 or AccuBlock and visualization using the LF160 visualizer. Parasitic loads per mL of blood are indicated at the bottom of the Figures. NC: negative control; PC: positive control.

Exclusivity was tested using heparinized blood and FTA cards spiked with 5000 *L. mexicana* promastigotes/mL. No amplification was obtained, indicating exclusivity (Fig 4).

## Analytical sensitivity and specificity of LAMP in different types of support

To assess the analytical sensitivity of the PURE-LAMP assay, at least 20 replicates of fluid or DBS samples containing known concentrations of Sylvio X-10 and CL-Brener stocks were tested. For this, we incubated the LAMP reaction for 40 minutes, as recommended by the manufacturer, as well as for 50 minutes in order to find out if a longer incubation of the amplification step managed to improve its performance (Table 1). Parasite-free blood samples were also analysed as control for analytical specificity (Table 1).

In the case of using heparinized blood spiked with CL-Brener or Sylvio X-10, the retrieved analytical sensitivity was between 5 and 10 par. Eq./mL, respectively (Table 1).

DBS spots gave analytical sensitivities between 10 and 20 par. Eq./mL depending on the strain and support. However, Whatman 903 based DBS samples with no parasites gave false

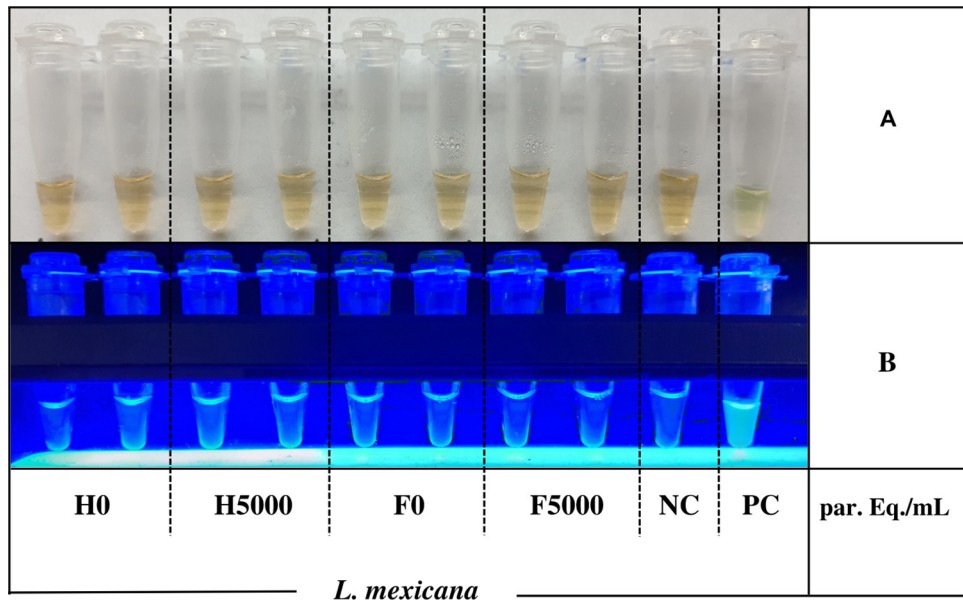

**Fig 4. FTA cards and heparinized blood containing 5000 *L. mexicana* promastigotes per mL to explore exclusivity of LAMP after PURE DNA preparation.** A) Naked eye; B) LF-160 light. Parasitic loads per mL of blood are indicated in the bottom of the Figure. F: FTA cards; H: heparinized blood; F0 and H0: non-spiked blood; NC: negative control; PC: positive control provided in the kit.

positive LAMP results in 25% to 40% of replicates, depending on the time of incubation (Table 1). The other procedures did not give rise to false positive results even when the LAMP reaction was incubated for 50 minutes. Thus, we discarded Whatman 903 as DBS support from further analysis.

## Evaluation of stability of DBS

We focused to carry on working with FTA cards as solid support. Stability was measured in FTA cards spiked with 20 par. Eq./mL of Sylvio X-10 stocks stored at room temperature for up to one year and a half after seeding the cards. Samples tested in duplicate during the first month, 2, 6, 12 and 18 months after preparation gave positive LAMP results. The FTA replicates tested at 12 and 18 months after preparation gave discordant negative and positive results. Thus, we repeated the experiment twice in duplicates, obtaining in total four positives out of six and three positives out of six LAMP results for each mentioned time-point, respectively, suggesting a diminution in sensitivity of FTA cards containing parasitic load close to the LoD concentration (Table 2).

## Discussion

The LAMP technique has been adapted for use in the diagnosis of different infectious diseases with notoriously high burdens, including tuberculosis and malaria, for which high accuracy and feasibility for POC are highly desired [11,14]. Similarly, it has been described it could be of major relevance for the detection of acute *T. cruzi* infections like those found upon congenital transmission of the parasite or during oral outbreaks [5–8].

Thinking of a versatile POC use of Eiken *T. cruzi*-LAMP prototype, this study encompassed the laboratory evaluation of FTA cards and Whatman 903 filter paper spiked with non-coagulated fresh blood in comparison with fluid heparinized blood to detect *T. cruzi* DNA using the

**Table 2. Stability of FTA cards at room temperature for LAMP tests.**

| Time (month) | Pos/Tested | Positivity |
|---|---|---|
| 1 | 2/2 | 100% |
| 2 | 2/2 | 100% |
| 6 | 2/2 | 100% |
| 12 | 4/6 | 67% |
| 18 | 3/6 | 50% |

FTA cards containing blood spiked with 20 par. Eq./mL of Sylvio X-10 stock and stored at room temperature for up to eighteen months after sample collection.

PURE system, also by Eiken, for rapid DNA extraction for downstream LAMP analysis. Because blood samples from suspected patients are collected directly in filter papers in the absence of anticoagulants, we decided to make the different spiked blood preparations with freshly drawn blood, which was artificially inoculated with known amounts of cultured parasites and taking all precautions to avoid coagulation.

We tested two types of filter paper supports (Whatman 903 and FTA) to find out which one was the most adequate for the LAMP reaction, considering that both have distinct characteristics and can be purchased in Chagas disease endemic countries. However, we discarded the subsequent use of Whatman 903 after realizing that parasite-free DBS samples extracted using the PURE system gave false positive results in a proportion of tested DBSs (Table 1). As FTA cards gave always non-detectable LAMP findings using the same non-spiked blood samples, putative laboratory contamination of the non-spiked blood with parasite DNA was discarded. Accordingly, we continued working with FTA cards. These consist of chemically modified filter paper that allow blood samples to be stored at room temperature because it lyses cells, inhibits the growth of microorganisms, denatures proteins, and immobilizes nucleic acids to the filter paper, protecting them from oxidation, the action of nucleases, and UV rays [15]. In addition, FTA cards have been used for the preservation and transport of different clinical samples from patients with vector-borne diseases [11,16]. In our hands, *T. cruzi* DNA was shown to be stable on FTA cards for at least six months at room temperature, which aligns with previous studies reporting successful pathogen isolation from cards stored under similar conditions [17]. Samples tested more than one year after preparation showed a diminution in sensitivity (Table 2). Published observations indicate that longer-term stability can be achieved when FTA cards are stored at 4°C, which remains to be tested using the PURE-LAMP technique [17].

It is well established that molecular methods yield higher sensitivity outcomes than the use of microscopy-based direct observation (e.g., micromethod) for the detection of *T. cruzi* congenitally infected infants [18]. Recently, a TaqMan qPCR-based kit including an internal control of DNA integrity of the sample or reaction inhibition (a.k.a. internal amplification control or IAC), further confirmed that, starting from 1 mL of peripheral blood mixed with a DNA stabilizer solution [19]. Under these conditions, qPCR sensitivity was around 1 par. Eq./mL, when tested on 5 μL purified DNA extracted from 300 μL from that stabilized blood sample [19].

Formerly, PCR has been used for the detection of *T. cruzi* DNA in DBS samples [6,20]. Nevertheless, to our knowledge, this is the first report of LAMP detection of *T. cruzi* DNA in DBS using an ultra-rapid DNA extraction method, specifically designed for POC diagnosis. A similar strategy has been already reported for the diagnosis of malaria [11]. Such a POC approach would mark the difference in resource-limited settings without the infrastructure to perform qPCR. PURE-LAMP would be an extremely helpful technique, being molecular-based, to

greatly increase the sensitivity of the current early diagnostic algorithm of congenital Chagas disease, which relies on suboptimal microscopy-based observations [18]. Blood loaded on FTA cards can be easily obtained at sites where a seropositive mother has delivered a baby, such as primary health posts or even mothers´ domiciles and transport it at room temperature to regional laboratories capable of performing LAMP tests but unable to run PCR. This is because the establishment of LAMP performance in a lab compared to that of performing PCR requires less investment, as well as easier and more feasible staff training and operating times.

In our experience, the PURE extraction procedure takes around 15 minutes if a single sample is processed or around 40 minutes when 14 samples are processed, number that represents the maximum quantity of samples that can be tested simultaneously in the LF-160 device involving two LAMP strips. Then, another 40 minutes are needed for LAMP incubation and 5 more minutes to stop the reaction and observe the results. In total, this would top around an hour and a half work to get to a molecular qualitative result. The output of LAMP can be visualized by the naked eye, using different acrylic sheets in a P51 box, or using the blue LED lightning component of the LF-160 incubator (Fig 2).

On the other side, at present, the estimate cost of the whole procedure would be around 15 USD per sample (3.3 USD for PURE and 11.5 for LAMP for research use in Argentina), whereas the cost for a commercial DNA extraction method coupled to a Chagas qPCR test assay would cost at least 25 USD.

In summary, the combination of PURE DNA isolation plus LAMP allowed adequate analytical sensitivity starting from only 30 μL of heparinized blood or a single 6-mm punch from a DBS collected in FTA paper. Heparinized blood allowed the detection of samples containing around 5 par. Eq./mL and FTA cards could potentially detect *T. cruzi* at concentrations around 15–20 parasites/mL in DBS, out of which 6-mm disc punches rendered the best performance in our hands. Such minimal loss of sensitivity between liquid and dried blood could be due to the fact that the sample volume is 10 μL, i.e. 3 times lower on the latter [21]. Nevertheless, this degree of analytical sensitivity is good enough to detect *T. cruzi* in most samples from congenitally or orally infected acute cases [18,19,22], that may be undetectable using current parasitological methods, which sensitivity is highly operator-dependent and it varies from 40 up to 1000 parasites/mL [18,23,24]. Overall, this work is a first step towards the design of operational field studies to validate the use of FTA sampling format for the rapid diagnosis of congenital Chagas disease in laboratories associated to Maternity Services in endemic and non-endemic regions as well as for rapid screening of infected cases in oral outbreaks.

## Supporting information

**S1 Table. PURE-LAMP step-by-step protocol.**
(PDF)

## Acknowledgments

We want to thank Dr Yasuyoshi Mori, Dr Shota Koyano, Dr Shindome Tsuyoshi and Dr Hosaka Norimitsu from Eiken Chemical Company, Tokio, Japan, for providing LF160 incubator, PURE and LAMP kits for fluid blood related work and technical assistance, and Dr Season Wong from AI BioSciences (College Station, TX, USA) for providing the P51 viewer (minipcr bio, Cambridge, MA, USA).

## Author Contributions

**Conceptualization:** Julio Alonso-Padilla, Alejandro G. Schijman.

**Data curation:** Silvia A. Longhi, Lady J. García Casares, Alejandro G. Schijman.

**Formal analysis:** Silvia A. Longhi, Julio Alonso-Padilla.

**Funding acquisition:** Julio Alonso-Padilla, Alejandro G. Schijman.

**Investigation:** Silvia A. Longhi, Julio Alonso-Padilla, Alejandro G. Schijman.

**Methodology:** Silvia A. Longhi, Lady J. García Casares, Arturo A. Muñoz-Calderón.

**Project administration:** Alejandro G. Schijman.

**Supervision:** Alejandro G. Schijman.

**Validation:** Silvia A. Longhi, Lady J. García Casares.

**Visualization:** Silvia A. Longhi, Lady J. García Casares, Arturo A. Muñoz-Calderón.

**Writing – original draft:** Silvia A. Longhi, Alejandro G. Schijman.

**Writing – review & editing:** Julio Alonso-Padilla, Alejandro G. Schijman.

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
