## [Decision Letter · Decision Letter 0]

9 Feb 2023

Dear Dr Schijman,

Thank you very much for submitting your manuscript "Combination of ultra-rapid DNA purification (PURE) and loop-mediated isothermal amplification (LAMP) for rapid detection of Trypanosoma cruzi DNA in dried blood spots." for consideration at PLOS Neglected Tropical Diseases. As with all papers reviewed by the journal, your manuscript was reviewed by members of the editorial board and by several independent reviewers. The reviewers appreciated the attention to an important topic. Based on the reviews, we are likely to accept this manuscript for publication, providing that you modify the manuscript according to the review recommendations that should be fully addressed. 

Sincerely,

Andrea Angheben

Academic Editor

Charles Jaffe

Section Editor

Reviewer's Responses to Questions

**Key Review Criteria Required for Acceptance?**

**Methods**

-Are the objectives of the study clearly articulated with a clear testable hypothesis stated?

-Is the study design appropriate to address the stated objectives?

-Is the population clearly described and appropriate for the hypothesis being tested?

-Is the sample size sufficient to ensure adequate power to address the hypothesis being tested?

-Were correct statistical analysis used to support conclusions?

-Are there concerns about ethical or regulatory requirements being met?

Reviewer #1: The parasitemia in congenital Chagas disease could be less than 1 parasite equivalents/mL, the authors assessed LOD up to 5 and 10 parasite equivalents/mL, why?

Why was the inclusivity just determined with 20 and 50 parasite equivalents/mL?

Reviewer #2: In a previous study of the group, published in 2017, the LAMP kit prototype (by Eiken, Japan) for qualitative detection of Trypanosoma cruzi DNA was evaluated for analytical sensitivity and specificity using peripheral human blood as clinical sample, in comparison to a standardized qPCR. For this, LAMP sensitivity was estimated using DNA extracted with commercial kits or after “Boil & Spin” rapid preparation. Now, in the present study, authors are interested in the standardization and analytical evaluation of the performance of this LAMP test, as a point-of-care (POC) molecular test to be used for the rapid diagnosis of congenital Chagas disease introducing FTA cards or Whatman 903 filter paper as solid supports for small-scale volumes of human blood. In addition, an ultrarapid purification system (PURE) for DNA extraction, also manufactured by Eiken, was evaluated by comparing artificially infected liquid blood in heparin as anticoagulant with different amounts of dried blood spot (DBS) either in FTA and Whatman 903 paper. A similar strategy using DBS coupled to PURE-LAMP as a point-of-care molecular test has been developed for malaria [Vincent et al. (2018) Combination of PURE-DNA extraction and LAMP-DNA amplification methods for accurate malaria diagnosis on dried blood spots. Malar J 17, 373]. 

The study design is appropriate to address the objectives.

There is no mention to Ethics statement regarding the use of seronegative human blood to obtain the spiked blood samples.

Minor issues:

DNA extraction using PURE system.

Considering that the study has a context of technological development for the implementation of a point-of-care diagnostic test and so that the standardized method can be replicated accurately, I suggest the authors to describe more clearly the step-by-step extraction by the PURE system, even if it is included as supplementary material.

Raised questions: 

Line 139. “mixed by shaking” – how many times, for how long (minutes)?

Lines 141-142. What is the approximate volume and concentration of the eluted DNA? Was the DNA measured before use?

Lines 143-145. What is the reason to add a minor volume of NaCl in higher volumes of blood compared to DNA extraction from 30 uL of blood? It should be related to the maximal capacity of the heating tube, 60 uL? The calculations to arrive at the final concentration of 10mM NaCl is not clear, considering the mixture of 30 ul of 334 mM NaCl + 30 ul blood.

Inclusivity and exclusivity.

Lines 172-174. Please, include “respectively” at the end of the sentence.

Analytical sensitivity and specificity.

Lines 184-186. Reference should be cited here [12 – Duffy et al., 2009] and also at the end of line 230 (Results).

**Results**

-Does the analysis presented match the analysis plan?

-Are the results clearly and completely presented?

-Are the figures (Tables, Images) of sufficient quality for clarity?

Reviewer #1: Line 205: The different experimental conditions should be explained in Methods section. Could be interesting to insert a paragraph about Study design.

Line 211: What were the results using 40 and 50 μL of blood?

Line 224: Figure 2 is not related to this part of the results.

line 226: In Methods, the authors indicate that inclusivity was evaluated using DNA from the 6 genotypes of T. cruzi, but they do not describe the results.

What were the results with T. rangeli?

Line 289: S1Table could be Table 2.

Line 310: What are the possible explanations for the false positives?

Line 322: “which remains to be tested in our case”, why?

Reviewer #2: The results and figures are clearly presented.

Line 208. To correct the volumes of heparinized blood used, as mentioned in methods - DNA extraction using PURE system. Tested volumes were 30 to 50 µL.

Line 218. What was the reason to add four times higher the volume of blood to the FTA cards compared to the 30 µL of dried blood included in each Whatman 903 spot? 

Table 1: The difference in the volume of blood between both DBS supports did not interfere in the positivity results?

Line 288. To include … obtaining in total four “positive” out of six LAMP results…

**Conclusions**

-Are the conclusions supported by the data presented?

-Are the limitations of analysis clearly described?

-Do the authors discuss how these data can be helpful to advance our understanding of the topic under study?

-Is public health relevance addressed?

Reviewer #1: The conclusion can be improved.

The limitations are not described.

Inclusion the real clinical samples could improve the relevance of this study.

Reviewer #2: The conclusions are very well supported and no limitations of analysis were described. Procedures were standardized for the LAMP reactions from small volumes of liquid blood or the use of FTA sampling format for the detection of T. cruzi DNA. The use of Whatman 903 filter paper was discarded due to the inespecificity of results (false positive). To validate the PURE-LAMP method in the field, in endemic and non-endemic regions, operational studies with neonates born to seropositive women are planned as well as the use of this point-of-care test for rapid screening of infected cases in oral Chagas disease outbreaks.

DISCUSSION.

Lines 320-322. A reference should be included.

Lines 362-364. The phrase is related to the work by Hewawasam et al. (2018). However, in the present study each DBS was blotted with 125 μL blood in Whatman FTA cards, 4 times higher than the tested volume for liquid heparinized blood. Could the authors comment please?

**Editorial and Data Presentation Modifications?**

Reviewer #1: (No Response)

Reviewer #2: This reviewer has no editorial suggestion. The revision includes minor modifications, as follows:

METHODS: DNA extraction using PURE system.

Considering that the study has a context of technological development for the implementation of a point-of-care diagnostic test and so that the standardized method can be replicated accurately, I suggest the authors to describe more clearly the step-by-step extraction by the PURE system, even if it is included as supplementary material.

Line 139. “mixed by shaking” – how many times, for how long (minutes)?

Lines 141-142. What is the approximate volume and concentration of the eluted DNA? Was the DNA measured before use?

Lines 143-145. What is the reason to add a minor volume of NaCl in higher volumes of blood, compared to DNA extraction from 30 uL of blood? It should be related to the maximal capacity of the heating tube, of 60 uL? The calculation to arrive at the final concentration of 10mM NaCl is not clear, considering the mixture of 30 ul of 334 mM NaCl + 30 ul blood.

Lines 172-174. Please, include “respectively” at the end of the sentence.

Lines 184-186. Reference should be cited here [12 – Duffy et al., 2009] and at the end of line 230 (Results).

There is no mention to Ethics statement regarding the use of seronegative human blood samples to obtain the spiked blood.

Line 208. To correct the volumes of heparinized blood used, as mentioned in methods - DNA extraction using PURE system. Tested volumes were 30 to 50 µL.

Line 218. What was the reason to add four times higher the volume of blood to the FTA cards compared to the 30 µL of dried blood included in each Whatman 903 spot? 

Table 1: The difference in the volume of blood between both DBS supports did not interfere in the positivity results between them?

Line 288. To include … obtaining in total four “positive” out of six LAMP results….

Lines 320-322. A reference should be included.

Lines 362-364. The phrase is related to the work by Hewawasam et al. (2018). However, in the present study each DBS was blotted with 125 μL blood in Whatman FTA cards, 4 times higher than the tested volume for liquid heparinized blood. Could the authors comment?

**Summary and General Comments**

Reviewer #1: The study describes the use of blood preservation on filter paper for its application in LAMP. But real clinical samples were not included. Since the limit of detection (LOD) of this methodology was 20 parasites/mL, lower parasitemias would not be detected. It would be convenient to point this out.

Reviewer #2: The group is strong and experienced in the development of molecular tests for the detection of T. cruzi and lately has been working on improving the LAMP methodology as a point-of-care diagnostic tool for the early diagnosis of acute infections such as congenital Chagas disease. Overall the study is very well designed with clear presentation of the results and it is original from the point of view of improving the LAMP test for application in the point-of-care diagnostic of acute T. cruzi infections. In this sense, the use of FTA cards as solid support for small-scale volumes of human blood is imperative for the preservation and transport of clinical samples from patients with acute Chagas disease. Another novelty of the study refers to the standardization of the PURE system for DNA extraction developed by Eiken, Japan, which reasonably decreases the time for DNA obtention compared to the use of specific columns provided in commercial kits and which require several steps until the DNA elution. 

It has to be highlighted the need to include Ethics statement for the use of human blood to obtain the spiked blood samples.

PLOS authors have the option to publish the peer review history of their article (what does this mean?). If published, this will include your full peer review and any attached files.

Reviewer #1: No

Reviewer #2: Yes: Constança Britto

Figure Files:

Data Requirements:

Reproducibility:

References

---

## [Decision Letter · Decision Letter 1]

6 Apr 2023

Dear Dr Schijman,

We are pleased to inform you that your manuscript 'Combination of ultra-rapid DNA purification (PURE) and loop-mediated isothermal amplification (LAMP) for rapid detection of Trypanosoma cruzi DNA in dried blood spots.' has been provisionally accepted for publication in PLOS Neglected Tropical Diseases.

Best regards,

Charles L. Jaffe, Ph.D.

Section Editor

Charles Jaffe

Section Editor

Reviewer's Responses to Questions

**Key Review Criteria Required for Acceptance?**

**Methods**

-Are the objectives of the study clearly articulated with a clear testable hypothesis stated?

-Is the study design appropriate to address the stated objectives?

-Is the population clearly described and appropriate for the hypothesis being tested?

-Is the sample size sufficient to ensure adequate power to address the hypothesis being tested?

-Were correct statistical analysis used to support conclusions?

-Are there concerns about ethical or regulatory requirements being met?

Reviewer #1: Everything is OK.

Reviewer #2: As requested, in the Methods section of the revised manuscript, a paragraph on "Ethics statement" was included.

Following that, another paragraph related to the “Study design” and a new figure (figure 1) were introduced. The new text and figure greatly facilitate the understanding of the technical aspects related to the methodology employed regarding the different tested conditions of sampling, incubation and visualization of LAMP results.

Also, for more clarity, the authors added a new supplementary table (S1 Table) in the revised version of the manuscript containing a step-by-step description of the DNA extraction procedure by the PURE system.

**Results**

-Does the analysis presented match the analysis plan?

-Are the results clearly and completely presented?

-Are the figures (Tables, Images) of sufficient quality for clarity?

Reviewer #1: Everything is OK.

Reviewer #2: Results and figures are clearly presented.

However, there is an error in Table 1; it is incomplete. Compared to the one presented in the first version of the manuscript, the two columns on the left are missing (Type of sample and Sampling conditions). Please, correct it.

Discussion - Line 348: please change Table 2 for Table 1.

**Conclusions**

-Are the conclusions supported by the data presented?

-Are the limitations of analysis clearly described?

-Do the authors discuss how these data can be helpful to advance our understanding of the topic under study?

-Is public health relevance addressed?

Reviewer #1: Everything is OK.

Reviewer #2: The conclusions are well supported and no limitations of analysis were described.

The analytical sensitivity of the PURE-LAMP protocol is sufficient to detect Trypanosoma cruzi DNA in samples from congenitally or orally infected acute cases and was shown to be higher than that reached by current parasitological methods. The next step is to validate the PURE-LAMP protocol with the FTA sampling format in field studies, for the rapid diagnosis of congenital Chagas disease and for the screening of infected cases in oral outbreaks.

**Editorial and Data Presentation Modifications?**

Reviewer #1: Accept.

Reviewer #2: No suggestions. Minor modifications were referred above.

**Summary and General Comments**

Reviewer #1: The manuscript has been substantially improved and shows the potential for the use of samples collected on filter paper for the molecular diagnosis of T. cruzi infection.

Reviewer #2: Overall, the revised manuscript presents a new and clear description of the study design, facilitating the understanding of the different experimental conditions and comparisons, beyond the inclusion of a step-by-step protocol for the new PURE system for DNA extraction.

PLOS authors have the option to publish the peer review history of their article (what does this mean?). If published, this will include your full peer review and any attached files.

Reviewer #1: No

Reviewer #2: **Yes: **Constança Britto

---

## [Editor Report · Acceptance letter]

10 Apr 2023

Dear Dr Schijman,

We are delighted to inform you that your manuscript, "Combination of ultra-rapid DNA purification (PURE) and loop-mediated isothermal amplification (LAMP) for rapid detection of Trypanosoma cruzi DNA in dried blood spots.," has been formally accepted for publication in PLOS Neglected Tropical Diseases.

Best regards,

Shaden Kamhawi

co-Editor-in-Chief

Paul Brindley

co-Editor-in-Chief
